# Switching the Local Symmetry from $D_{5h}$ to $D_{4h}$ for Single-Molecule Magnets by Non-Coordinating Solvents

Xia-Li Ding [1], Qian-Cheng Luo [1], Yuan-Qi Zhai [1], Qian Zhang [2,*], Lei Tian [3], Xinliang Zhang [4], Chao Ke [5], Xu-Feng Zhang [6], Yi Lv [6] and Yan-Zhen Zheng [1,*]

[1] Frontier Institute of Science and Technology (FIST), Xi'an Jiaotong University Shenzhen Research School, State Key Laboratory for Mechanical Behavior of Materials, MOE Key Laboratory for Nonequilibrium Synthesis and Modulation of Condensed Matter, Xi'an Key Laboratory of Sustainable Energy and Materials Chemistry, School of Chemistry and School of Physics, 99 Yanxiang Road, Xi'an 710054, China; dxl1457337646@163.com (X.-L.D.); luoqiancheng@stu.xjtu.edu.cn (Q.-C.L.); zhaiyuanqi@stu.xjtu.edu.cn (Y.-Q.Z.)

[2] State Key Laboratory of Military Stomatology & National Clinical Research Center for Oral Diseases & Shaanxi International Joint Research Center for Oral Diseases, Department of General Dentistry and Emergency, School of Stomatology, Air Force Medical University, 169 West Changle Road, Xi'an 710032, China

[3] State Key Laboratory of Military Stomatology, National Clinical Research Center of Oral Diseases, Shaanxi Key Laboratory of Oral Diseases, Department of Cranio-facial Trauma and Orthognathic Surgery, School of Stomatology, The Fourth Military Medical University, 145 West Changle Road, Xi'an 710032, China; tianleison@163.com

[4] Department of Spine Surgery, Honghui Hospital, Xi'an Jiaotong University, 76 Nanguo Road, Xi'an 710054, China; xaxinliang@stu.xjtu.edu.cn

[5] Department of Orthopaedic Trauma, Honghui Hospital, College of Medicine, Xi'an Jiaotong University, 28 West Xianning Road, Xi'an 710054, China; kechao373855288@163.com

[6] Department of Hepatobiliary Surgery, The First Affiliated Hospital of Xi'an Jiaotong University, 277 West Yanta Road, Xi'an 710061, China; xfzhang125@126.com (X.-F.Z.); luyi169@126.com (Y.L.)

* Correspondence: laolvshu@126.com (Q.Z.); zheng.yanzhen@xjtu.edu.cn (Y.-Z.Z.); Tel.: +86-029-833-951-72 (Y.-Z.Z.)

**Abstract:** A solvent effect towards the performance of two single-molecule magnets (SMMs) was observed. The tetrahydrofuran and toluene solvents can switch the equatorial coordinated 4-Phenylpyridine (4-PhPy) molecules from five to four, respectively, in $[Dy(O^tBu)_2(4\text{-}PhPy)_5]BPh_4$ **1** and $Na\{[Dy(O^tBu)_2(4\text{-}PhPy)_4][BPh_4]_2\}\cdot2thf\cdot hex$ **2**. This alternation significantly changes the local coordination symmetry of the Dy(III) center from $D_{5h}$ to $D_{4h}$ for **1** and **2**, seperately. Magnetic studies show that the magnetic anisotropy energy barrier of **2** is higher than that of **1**, while the relation of blocking temperature is just on the contrary due to the symmetry effect. The calculations of the electrostatic potential successfully explained the driving force of solvents for the molecular structure change, confirming the feasibility of adjusting the performance of SMMs via diverse solvents.

**Keywords:** dysprosium; local symmetry; single-molecule magnets; magnetic relaxation; solvent effect

## 1. Introduction

There are considerable high performance single-molecule magnets (SMMs) containing dysprosium(III) reported in recent years [1–6], ranging from the cyclopentadienyl (Cp) based system [7–10] to the pentagonal-bipyramidal (PB) family [11–14]. Among these dysprosium(III)-based complexes, highly axial crystal field [14,15] is essential to achieve good performance of SMMs [16,17]. In this regard, other ancillary components, while unnecessary to this target, play, however, a key role to stabilize the Dy(III) complexes because the radii of the lanthanides are very large [3,4,18]. This is well demonstrated by Tong et al. who reported a text-book example of solvent induced symmetry transformation between quasi-$D_{5h}$ and quasi-$O_h$ by losing its one coordinated methanol molecule when exposed to the dry air and then recovered the structure by soaking in methanol for one

day [19]. The compounds with identical ligands are provided with completely different magnetic properties, as well as the effective energy barrier ($U_{eff}$) and blocking temperature ($T_B$), which are two pivotal indexes evaluating the properties of SMMs [20–22].

Furthermore, it has also been discovered that the similar transform in configuration triggered by the solvent effect leads to the change of related properties, such as hydrophobicity and luminescence response [23–25]. Zang's group synthetized one dendrimers of Ag$_{12}$@POSS$_6$ by introducing polyhedral oligomeric silsesquioxane (POSS) with thiol group modifying and observed that its core cluster went through a structural change within flattened cubo-octahedral and normal cubo-octahedral caused by different solvents of acetone and tetrahydrofuran, making the film matrix embellished by them possess distinctive hydrophobicity [23]. Zhang et al. prepared one case of dumbbell-shaped crystalline molecular rotor and they found that it and its solvated crystal possess a structural difference and that a dihedral angle change of about 30° exists, leading to a luminescent change of about 10 nm [24].

However, other than the direct coordinating solvent, here, we show that the uncoordinated solvent can also have a significant impact on the final coordination number of the equatorial ligands of two dysprosium(III) SMMs. Using the similar synthetic procedure with only varying the reaction solvents, we observed that the tetrahydrofuran (THF) and toluene solvents can switch the equatorial coordinated 4-Phenylpyridine (4-PhPy) molecules from five to four, which finally leads to two complexes, namely, [Dy(O$^t$Bu)$_2$(4-PhPy)$_5$]BPh$_4$ **1** and Na{[Dy(O$^t$Bu)$_2$(4-PhPy)$_4$][BPh$_4$]$_2$}·2thf·hex **2**. The corresponding point group of the molecule is altered from local $D_{5h}$ to $D_{4h}$, respectively. It should be noted that the PB complexes are usually crystallized in THF or pyridine solvents and this is the first time that we isolated the PB complex in toluene. We have calculated the electrostatic potentials (ESPs) of the equatorial ligand and two classes of solvents, from which we could see the configuration transition in different solvent was validated due to the varied electrostatic interaction between the solvent and ligand. Considering the different properties of both SMMs, this variety of supramolecular chemistry process realizes the switch of magnetic control of compounds.

## 2. Results and Discussion

### 2.1. Synthesis of 1–2

The synthesis steps of these two complexes are shown in Scheme 1. The only difference is the treatments of the powder after being dried from the previous reaction using the THF solvent.

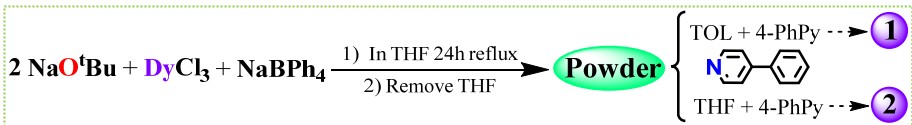

**Scheme 1.** The synthetic route for Complex **1** and **2**.

### 2.2. Single Crystal Structure

The X-ray single crystal structure analysis reveals that the crystallographic asymmetric unit of **1** is composed of a seven-coordinate mononuclear cation [Dy(O$^t$Bu)$_2$(4-PhPy)$_5$]$^+$ and a charge-balancing counteranion BPh$_4^-$ (Table S1). The central Dy(III) center in the asymmetric unit of **2** is six-coordinate with the formula of [Dy(O$^t$Bu)$_2$(4-PhPy)$_4$]$^+$ (Figure 1). The structure of **1** is very similar to previously reported dysprosium(III) complexes belonging to the family of PB geometry. The average Dy-O distance of 2.123(2) Å for **1** is a little longer than that of **2** (2.066(8) Å); the five equatorial Dy-N bond lengths are ranging from 2.55 to 2.59 Å for **1**, which are obviously longer than those of **2** (averaged at 2.468(8) Å, see Table S2 for detail). Indeed, the PB polyhedron is quite regular with an axial O-Dy-O angle of 175.14(7)° and 180° for **1** and **2**, respectively, and N-Dy-N angles ranging from 69.44(12)° to 73.41(11)°. These geometries lead to the continuous shape measurement (CShM) calculations [26] for the Dy$^{III}$ ions of 0.889 for local $D_{5h}$ symmetry of **1** and 0.693 for the

compressed octahedron of **2**, namely with tetragonal–bipyramidal ($D_{4h}$) local-symmetry (Table S3).

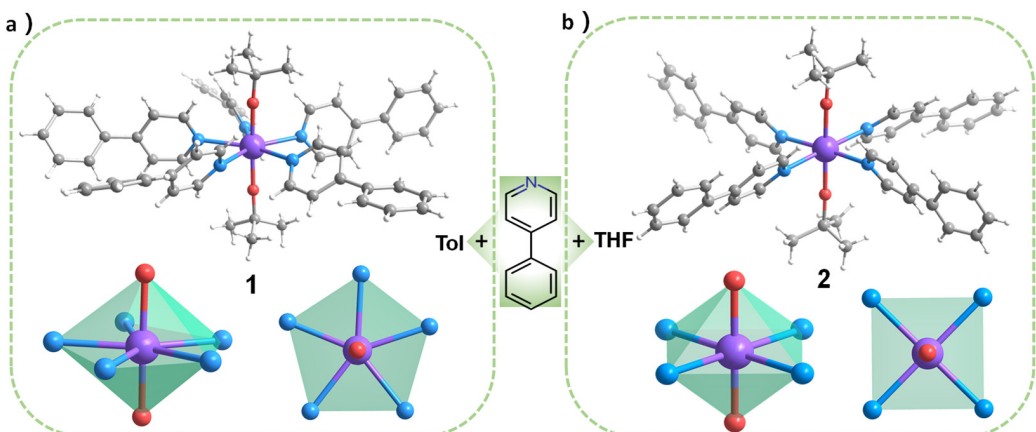

**Figure 1.** The molecular structures of the [Dy(O$^t$Bu)$_2$(4-PhPy)$_5$]$^+$ cation in **1** (**a**) and [Dy(O$^t$Bu)$_2$(4-PhPy)$_4$]$^+$ cation in **2** (**b**); other atoms and disordered two THF molecules, one hexane molecule and one Na$^+$ cation per formula unit in the channels are omitted for clarity in **2**. The polyhedron of the Dy$^{III}$ ions center viewed respectively from the front (left) and the top (right).

In the molecular packing, the shortest intermolecular Dy···Dy distance is 12.57(3) Å for **1** and 15.63(2) Å for **2** (Figures S2 and S3), respectively. More interestingly, the disordered organic THF/hexane molecules are easily lost in the 3D extended channels structure formed by the intermolecular π-π stacking interactions (Figure 2a–c), with a 2/3 voids existing in the 3D extended channels structure of compound **2** according to the 'SQUEEZE' routine. Further characterizations indicate that there are two disordered THF molecules, one disordered hexane molecule and one Na$^+$ cation per formula unit in the channels as confirmed by ICP, TGA, elemental analysis, and charge balance (details can be seen in the previous report [5]). There is no such special supermolecular structure for **1** (Figure 2d). However, the solvent is readily lost in **2**, at a temperature in THF with no structural changes, only losing the guest molecules while the host structure was maintained. We did not find the structure of compound **2** to revert to compound **1** in the process of measurement. It is possible that the solubility of Na[BPh$_4$] in THF vs. toluene is a large driver of the change in structure. This also manifests that solvation has an important influence on the coordination geometry and spatial packing structure of the complexes, which probably lead to having different effects on the magnetic properties of the complexes.

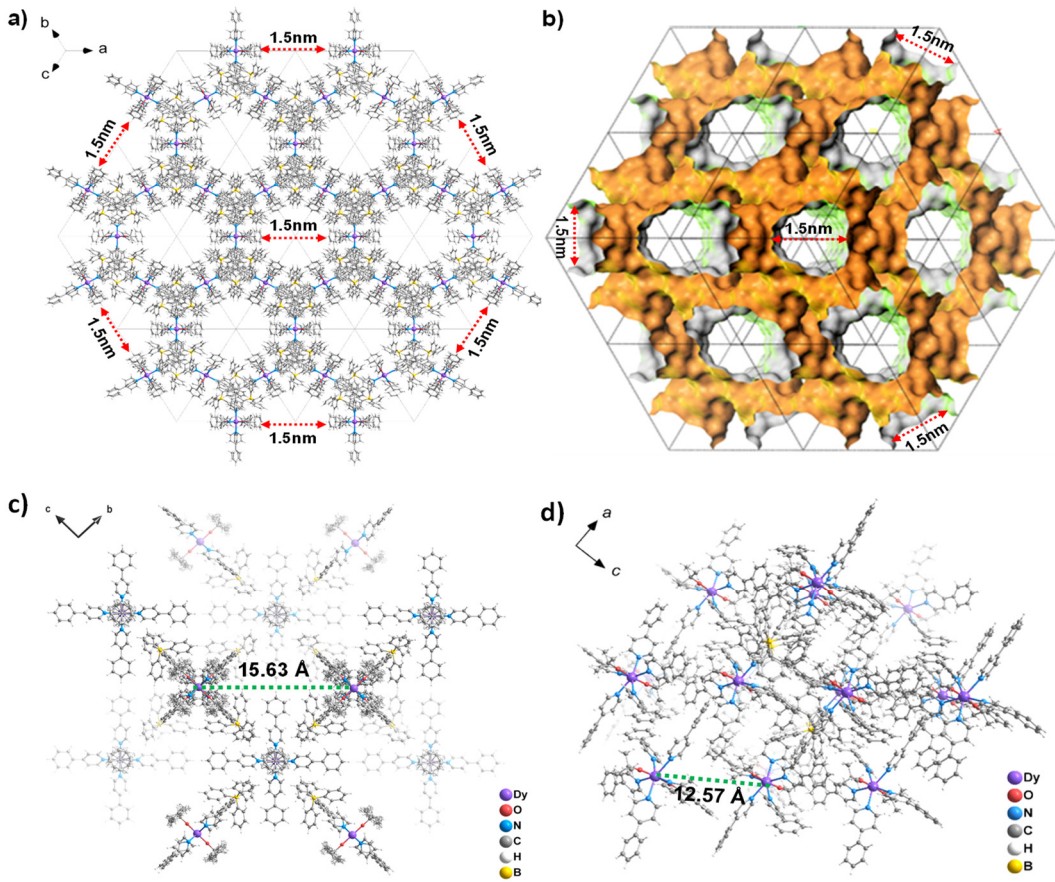

**Figure 2.** (**a**) The packing diagram of the main skeleton structure for **2** in one asymmetric unit cell. (**b**) The 3D extended channels structure for **2**; the BPh$_4$ and disordered components in channels are omitted for clarity. (**c**) The packing diagram for **2** along the a direction. (**d**) The packing diagram for **1** along the a direction.

*2.3. Magnetic Properties*

From the temperature-dependent direct current (dc) magnetic susceptibilities, complexes **1** and **2** show similar paramagnetism. The temperature dependence of the magnetic susceptibilities were carried out under 1 kOe dc field in the temperature range of 300–2 K (Figure S4), which gave the $\chi T$ products (in the unit of cm$^3$ K mol$^{-1}$) of 14.08 for **1** at 300 K, which are very close to the expected value of 14.17 cm$^3$ K mol$^{-1}$ for free Dy$^{3+}$ ion. Upon cooling, $\chi T$ values in two complexes keep essentially constant. At lower temperature about 20 K, the sudden drop of $\chi T$ product indicates the onset of magnetic blocking (Figure S4).

The field (*H*) dependence of the magnetization (*M*) for **1** (Figure 3a) were measured at a 10 Oe/s sweeping rate. For **1**, the magnetic hysteresis loops were not closed up to 13 K, which are in substantial agreement with the zero-field-cooled and field-cooled (ZFC-FC) magnetizations peak at about 11 K for **1** (Figure 3c). On the other hand, the butterfly magnetic hysteresis loops were observed only up to 5 K and did not have a peak in ZFC-FC magnetizations for **2** as shown previously [5]. Moreover, we could see a smaller step in zero dc field regime and even a wider hysteresis loop at 2 K for **1** compared to **2** (Figure 3b), which indicates a blocking temperature of SMM with $D_{5h}$ local symmetry clearly higher than the one with $D_{4h}$ local symmetry. We reasoned that this is because the pseudo-$D_{5h}$ symmetric SMM can better restrain the quantum tunneling of magnetization (QTM) effect at zero field and hence, $T_B$ is enhanced.

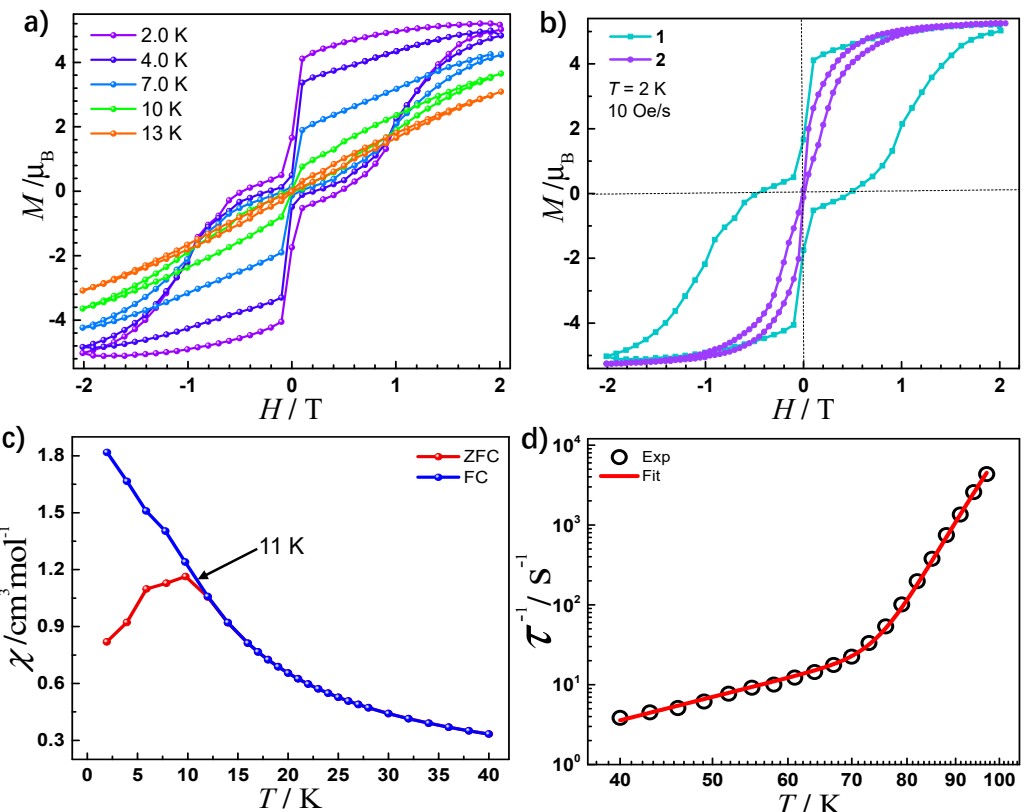

**Figure 3.** (**a**) Magnetic hysteresis loops for **1**. (**b**) The comparison of the magnetic hysteresis at 2 K for **1** and **2**. (**c**) Field-cooled (FC) and zero-field-cooled (ZFC) magnetic susceptibilities of **1** measured under a dc field of 1000 Oe. (**d**) Temperature dependence of the relaxation time $\tau$ in a zero dc field for **1**, the solid lines are the best fits according to the equation $\tau^{-1} = \tau_0^{-1}e^{-U_{eff}/T} + CT^n$.

Alternating current (ac) susceptibilities under zero dc field were measured for **1** (Figure S5 and S6) at a relatively high temperature range, and for **2** in details, they can be seen in the previous report [5]. The maxima of both the in-phase ($\chi'$) and out-of-phase ($\chi''$) components showed a clear temperature dependence (Figure S5). The frequency-dependent ac susceptibilities were also measured and the peaks of $\chi''$ can be all observed from 40–97 K for **1** (Figure S6a) at the frequency ranging from 1 to 1218 Hz. Moreover, the frequency-dependent data can be by a modified Debye function to obtain the Cole−Cole plots for **1** (Figure S7). The $\alpha$ value is less than 0.01 (Table S4) and only has a peak, demonstrating narrow distributions of relaxation times and one relaxation mechanism for this sample. Figure 3d for **1** shows the plots of $\tau^{-1}$ versus T for the temperature-dependent relaxation rate. The $\tau$ values are obtained by fitting the ac data with the equation $\tau^{-1} = \tau_0^{-1}e^{-U_{eff}/T} + CT^n$, giving the following parameters: for **1**, $U_{eff} = 1785(6)$ K, $\tau_0 = 1.96(4) \times 10^{-12}$ s, $C = 5.3(6) \times 10^{-5}$ s$^{-1}$ K$^{-n}$, $n = 3.02(5)$. These parameters are very similar to other $D_{5h}$ SMMs due to the fact that they share the same coordination geometry [27]. For **2**, the equation has been modified with an additional QTM term, which gives $U_{eff} = 2075(11)$ K, $\tau_0 = 5.61(2) \times 10^{-13}$ s$^{-1}$, $C = 5.60(4) \times 10^{-3}$ s$^{-1}$ K$^{-n}$, $n = 2.85(4)$ and $\tau_{QTM} = 0.46$ s. From these results, we could see that in the higher temperature the relaxation mechanisms for both complexes are similar, whereas in the low temperature region where QTM is effective, the relaxation behavior is disparate for two complexes. Due to its local $D_{4h}$ symmetry effect, **2** has stronger QTM effect compared to **1**. For **1**, due to the nearly perfect $D_{5h}$ local symmetry, QTM is much suppressed. This can be also seen in the hysteresis loops (Figure 3). At 2 K (Figure 3b) at the zero field region, the hysteresis is much wider for **1** than that of **2**, indicating that QTM is well mitigated [28].

*2.4. Ab initio Calculation*

　　To gain insight into the magnetic properties and electronic structure, ab initio calculation towards **1** was performed at SA-CASSCF/RASSI level by using OPEN MOLCAS (see Supporting Information for details). The highly axial g-tensor of ground Kramer Doublet (KD) could be observed ($g_x = g_y = 0$, $g_z = 19.89$) as well as local principal magnetizations of that almost along the O-Dy-O direction (Figure 4, insert), and its wave function is rather pure with 99.9% $|\pm15/2\rangle$ (Table S5). Meanwhile, the calculated LoProp charges also prove the existence of a strong axial crystal field with more negative charges on O atoms than equatorial N atoms (Table S7). Moreover, the excited KDs contain relatively pure wave function until the fourth KD: 99.8% $|\pm13/2\rangle$ (KD$_2$), 99.4% $|\pm11/2\rangle$ (KD$_2$), 87.1% $|\pm9/2\rangle$ (KD$_3$) and 70.1% $|\pm1/2\rangle$ (KD$_4$) lying at 781 K, 1300 K, 1576 K and 1636 K, respectively. The third excited KD with highly mixed characteristic and its $g_z$ angle of 86.86° indicate that the Orbach relaxation passes through this state, making the theoretical value of $U_{eff}$ around 1736 K, which is closely consistent with the experimental value (Figure 4).

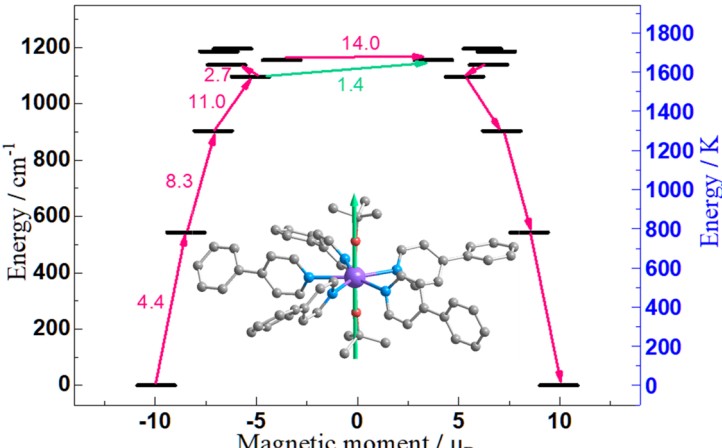

**Figure 4.** Ab initio calculated electronic states of **1**. The numbers beside the arrows express the relative transition propensity. The local principal magnetizations of the ground Kramer's doublet of **1**. For clarity, all hydrogen atoms are omitted (insert).

　　The interrelated complex **2** possessing local $D_{4h}$ symmetry has already been discussed via crystal field parameters (CFPs) in the previous literature, indicating that such geometry adverts the emergence of significant non-axial crystal field terms ($B_k^q$, $q \neq 0$) and can be one of the candidates to design high performance SMMs. Notwithstanding strong uniaxial crystal fields realized in both complexes, the $U_{eff}$ value for **1** is lower than that of **2**, which can be understood by means of the following aspects. Firstly, the increment of the coordination number in the equator circle partly causes the reduction of the local symmetric environment around Dy$^{3+}$ ions, which proceeds to changing CFPs. We found that the absolute value of axial parameter $B_2^0$ in **1** (−6.53) is less than that in **2** (−9.50), hinting identical relationship in respect of respective axiality of ground KD. Moreover, the $g_z$ value of ground KD in **1** (19.89) is also smaller compared with **2** (19.98). Furthermore, from the perspective of magneto-structural correlation, the length of Dy-O in **1** is larger than in **2** while the relationship reverses for Dy-N bonds ($L_{Dy-O(avg)}$ = 2.123 Å, $L_{Dy-N(avg)}$ = 2.577 Å for **1**, $L_{Dy-O(avg)}$ = 2.064 Å, $L_{Dy-N(avg)}$ = 2.648 Å for **2**). Evidently, the shorter Dy-O distance plays a leading role in such competitive effect between both different types of bond length. Meanwhile, the angle of O-Dy-O in the latter is 179.95°, more linear than **1** (175.14°), making this variety of coordination mode more suitable for ions with oblate electron density.

　　In addition, electrostatic potentials (ESPs) of distinctive solvents and transversal ligands were calculated to explain configuration transformation in distinctive solvents (Figure 5). It was found that THF and 4-PhPy possess similar electrostatic potential distri-

bution, while there was no or only a little negative potential region on the surface of toluene. Due to pure electrostatic attractions between Dy and N, in THF solvent environment, the molecules of THF and 4-PhPy are mutually exclusive in electrostatic interaction, leading to the formation of a complex with local $D_{4h}$ symmetry. Meanwhile, the rotatability of the carbon–carbon single bond between the two rings provides favorable conditions for the formation of this configuration in terms of steric hindrance. Contrarily, a compound with local $D_{5h}$ configuration is formed in toluene solvent owing to the fact that molecules tend to contact in a complementary way to maximize the electrostatic interaction. In short, from the nature of bonding, we understand the phenomenon of solvent-induced configuration transformation, which affects the properties of SMMs and it is the flexibility of supramolecular chemistry that leads to these interesting and meaningful coordination reactions emerging.

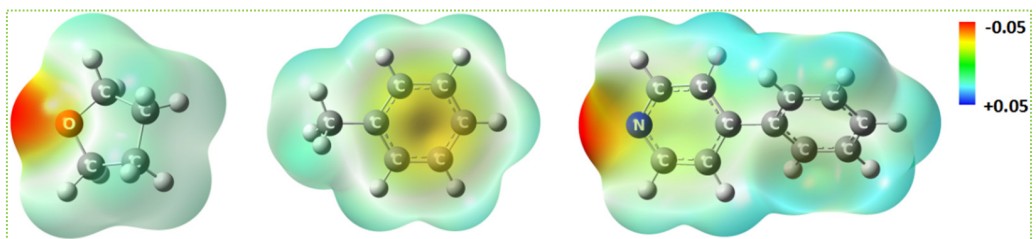

**Figure 5.** Electrostatic potentials of THF, toluene, and 4-PhPy from left to right, respectively. (isovalue = 0.001).

## 3. Materials and Methods

### 3.1. Synthesis

For **1**, in an argon glovebox, a mixture of DyCl₃ (0.5 mmol, 134 mg), NaO$^t$Bu (1 mmol, 96 mg) and NaBPh₄ (0.5 mmol, 171 mg) was added with about 10 mL THF in Schlenk tube, which gave a cloudy solution. After stirring for 24 h, the solution was filtered and the solvent was removed by vacuum to get a white powder of the products; 2 mL toluene and 4-PhPy (3 mmol, 465 mg) were then added to the powder with further stirring. Colorless crystals suitable for X-ray diffraction were grown by slow diffusion of hexane to the solution at room temperature after three days: Yield 278 mg, 36% (based on Dy); Elemental analysis calcd (%) for $C_{98}H_{95}BDyN_5O_2$: C 76.03, H 6.19, N 4.52; found: C 76.05, H 6.20, N 4.53. IR was as showed in Figure S1 (see Supplementary Materials).

The synthesis process of **2** is the same as **1** with THF used instead [5].

### 3.2. X-ray Crystallography Data

All data were recorded on a Bruker SMART CCD diffractometer with MoK$\alpha$ radiation ($\lambda$ = 0.71073 Å). The structures were solved by direct methods and refined on $F^2$ using Olex2. CCDC 2086927 (**1**) contains the supplementary crystallographic data for this paper including Tables S1–S4. The CIF and the checkCIF files for **1** can be found in the Supplementary Materials.

### 3.3. Magnetic Properties

Magnetic susceptibility measurements were carried out with a Quantum Design MPMS-XL7 SQUID magnetometer (Quantum Design Company, San Diego, CA, USA) upon cooling from 300 to 2 K in variable applied fields. Ac susceptibility measurements were performed at frequencies of between 1 and 1500 Hz with an oscillating field of 3.5 Oe and with variable dc applied field. Powder samples were embedded in eicosane to avoid any field induced crystal reorientation. Crystalline powders were fixed with eicosane, wrapped with film, and placed in the center of a straw. A diamagnetic correction has been calculated from Pascal constants and embedding eicosane has been applied to the observed magnetic susceptibility. The results are included in Table S4 and Figures S4–S7 (see Supplementary Materials).

### 3.4. Electronic Structure Calculations

Complete Active Space Self-Consistent Field (CASSCF) calculation was performed to understand the magnetic properties of **1** via its electronic structure using OPEN MOL-CAS [29] and its geometry structure was obtained straightly from X-Ray single crystal structure without optimization. The basis sets from the ANO-RCC library [30] were employed for all atoms: VTZP quality for Dy, VDZP quality for O and N atoms, as well as VDZ quality for others; 21 sextets, 224 quartets, and 490 doublets were calculated in the RASSCF module to acquire the state-averaged CASSCF orbitals. Then, a spin-orbit (SO) coupling Hamiltonian was constructed and diagonalized in the RASSI module [31] through the chosen 21 sextets, 128 quartets, and 130 doublets. Ultimately, all magnetic properties of Dy(III) ion, such as g-tensors, crystal field parameters, transition magnetic moment matrix, magnetic susceptibility, and magnetization plot, were computed and the output was obtained via the SINGLE_ANISO program [32]. For assuring calculation accuracy, we also considered employing the Cholesky decomposition for two-election integrals. Including Tables S5–S8 (see Supplementary Materials).

### 3.5. DFT Calculations

To acquire wave function information of both complexes and electrostatic potentials (ESP) of distinctive solvents and transversal ligand, the calculations based on Density Function Theory (DFT) were performed using Gaussian 09 E01 [33]. The PBE density functional [34] was employed in all calculations with Grimme's D3 dispersion correction considered [35–37]. Primarily, the positions of hydrogen atoms of **1** and **2** were optimized. We replaced Dy(III) ion to Y(III) in light of similar radius between them, and set its atomic mass as 162.5 (the same as natural abundance-weighted mass of dysprosium). The Stuttgart RSC 1997 effective core potential (ECP) [38,39] was applied for 28 core electrons of Y(III) and corresponding valence basis set was used for the remaining valence electrons, while the rest of atoms were treated with cc-pVDZ basis set [40,41]. Then the whole molecules of toluene, THF and 4-phenylpyridine were optimized by the same basis set. Harmonic vibrational calculations indicate that there is no imaginary vibration mode and all optimized minimum-energy structures have already been at stationary points on the potential energy surface. The results are included in Tables S9–S13 (see Supplementary Materials).

## 4. Conclusions

To summarize, we observed, for the first time, that a non-coordinating solvent can also significantly impact the coordinate number of the equatorial ligands in the pentagonal–bipyramidal geometry of the dysprosium(III) SMMs. The performances of SMMs are subsequently switched. The effective energy barrier in **2** is higher than **1** while the relation of blocking temperature is just the contrary. The structural transform is explained by the electrostatic interaction between the solvent and the ligands, which confirmed the feasibility of adjusting the performance of SMMs by supramolecular chemistry of solvents effect.

**Supplementary Materials:** The following are available online at https://www.mdpi.com/article/10.3390/inorganics9080064/s1, Figure S1: The IR spectrum of complex 1 and 2; Table S1: Crystallographic data for complex 1 and 2; Table S2: Selected Bond Lengths (Å) and Bond Angles (deg) for complex 1 and 2; Table S3: The CShM's values of the first coordination sphere of compound 1; Table S4: The CShM's values of the first coordination sphere of compound 2; Figure S2: Packing diagram for complex 1; Figure S3: Packing diagram for complex 2; Table S4: Relaxation fitting parameters of a generalized Debye model for 1; Figure S4: The variable-temperature dc magnetic susceptibility (1 kOe) and the field dependence of the magnetization (inset) at 2 K for 1; Figure S5: Temperature-dependence of the in-phase χ′ and out-of-phase χ″ ac susceptibility signals under zero dc field for 1; Figure S6: Frequency-dependence of the in-phase χ′ and out-of-phase χ″ in a zero dc field for 1 with ac frequencies of 1–1218 Hz; Figure S7: Cole-Cole plots for the ac susceptibilities in a zero dc field for 1; Table S5: Ab initio results for the J = 15/2 multiple of DyIII in 1; Table S6: Ab initio calculated crystal field parameters for 1; Table S7: Ab initio calculated LoProp charges of the atoms near Dy center in 1; Table S8: Average transition magnetic moment elements between the states

of 1; Table S9: Geometry coordinates of 1 in population analysis calculations; Table S10: Geometry coordinates of 2 in population analysis calculations; Table S11: Geometry coordinates of toluene molecule in population analysis calculations; Table S12: Geometry coordinates of THF molecule in population analysis calculations; Table S13: Geometry coordinates of 4-phenylpyridine in population analysis calculations. The CIF and the checkCIF files for **1**.

**Author Contributions:** Conceptualization, X.-L.D and Y.-Z.Z.; methodology, X.-L.D., Q.-C.L., Y.-Q.Z. and Y.-Z.Z.; software, X.-L.D., Q.-C.L. and Y.-Q.Z.; validation, X.-L.D., Q.-C.L., Y.-Q.Z., Q.Z., L.T., C.K., X.Z., X.-F.Z., Y.L. and Y.-Z.Z; formal analysis, X.-L.D.; Y.-Q.Z. and Q.-C.L.; investigation, Y.-Z.Z., Q.Z., L.T., C.K., X.Z., X.-F.Z. and Y.L.; resources, Q.Z., L.T., C.K., X.Z., X.-F.Z. and Y.L.; data curation, X.-L.D., Q.-C.L. and Y.-Q.Z.; writing—original draft preparation, X.-L.D. and Q.-C.L.; writing—review and editing, X.-L.D., Q.-C.L. and Y.-Z.Z.; visualization, Y.-Z.Z.; supervision, Y.-Z.Z.; project administration, Y.-Z.Z.; funding acquisition, Y.-Z.Z., Q.Z., L.T., C.K., X.Z., X.-F.Z. and Y.L. All authors have read and agreed to the published version of the manuscript.

**Funding:** This research was funded by the National Natural Science Foundation of China (21871219, 21773130 and 21620102002); Key Laboratory Construction Program of Xi'an Municipal Bureau of Science and Technology (201805056ZD7CG40); The Shenzhen Science and Technology Program (JCYJ20180306170859634); The Fundamental Research Funds for Central Universities.

**Data Availability Statement:** All data needed to evaluate the paper are present in the main text or the Supplementary Materials. Additional data related to this paper may be requested from the authors.

**Acknowledgments:** This work was supported by the following funding and authors. We also thank the Instrument Analysis Center of Xi'an Jiaotong University for the assistance from Gang Chang.

**Conflicts of Interest:** The authors declare no conflict of interest.

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
