# Peer review of "Switching the Local Symmetry from D5h to D4h for Single-Molecule Magnets by Non-Coordinating Solvents"

_inorganics, doi:10.3390/inorganics9080064_

Round 1
Reviewer 1 Report
The paper by Yan-Zhen Zheng and co-workers provides a thorough investigation of a new complex [Dy(OtBu)2(4-PhPy)5][BPh4] (1) with single-molecule magnet (SMM) behaviour, and draws direct comparisons to their previously reported compound [Dy(OtBu)2(4-PhPy)4][BPh4] (2). Although the synthetic procedures are identical apart from the solvent employed (toluene for 1 and THF for 2), this difference results in changes in the coordination environment around the Dy centre (7-coordinate for 1 and 6-coordinate for 2), despite the solvent being non-coordinating. This change in coordination number causes a change in symmetry around the Dy centre, which in turn has consequences on the magnetic properties. For 1, where more ligands are in the equatorial plane, symmetry is lowered and thus the barrier to relaxation is smaller than that of 2, which possesses higher symmetry around the Dy centre. Through structural and detailed magnetic investigations, supported by ab initio calculations, Yan-Zhen Zheng and co-workers draw on magneto-structural correlations to explain the differences in the SMM properties of 1 and 2. They also make use of theoretical calculations to determine the role of the solvent in determining the coordination environment, which was postulated to arise due to the differences in the electrostatic potentials of THF versus toluene, and as a result how these interact with the 4-PhPy ligands which reside in the equatorial positions.
The review is very thorough and well written; it will certainly be of interest to the journals readers. However, I have a few minor suggestions to improve the manuscript:
- In abstract the authors write ‘More interestingly, this process is reversible and the magnetic energy barrier of 2 is higher than that of 1; while the relation of blocking temperature is just on the contrary due to the symmetry effect.’ I am not sure in which way the coordination number change is reversible, so this should be clarified in the main text.
- The environment around the Dy centres is not strict in terms of symmetries, and so it is more correct to be referred to as pseudo-D5h and D4h OR add a sentence in the crystallography section to explain that the symmetry is approximate and not strictly D5h and D4h.
- The authors refer a lot to their previous report of complex 2, but don’t provide a reference for this other in the introduction where it is included in reference to previous Dy SMMs along with multiple others. It would certainly help the reader if this reference could be added where previous data (e.g. crystallography and magnetic properties) is referred to.
- Sentence starting line 88: ‘The X-ray single crystal structure analysis reveals that the two complexes respectively compose of mononuclear seven-coordinate cation with [Dy(OtBu)2(4-PhPy)5]+ for 1 and six-coordinate cation with [Dy(OtBu)2(4-PhPy)4]+ for 2 motif’. The way this is phrased implies both complexes are seven coordinate. I would suggest the authors rephrase along the lines of ‘The X-ray single crystal structure analysis reveals that complex 1 is composed as a mononuclear mononuclear seven-coordinate cation with [Dy(OtBu)2(4-PhPy)5]+, whereas complex 2 has a six-coordinate cation motif [Dy(OtBu)2(4-PhPy)4]+’.
- Line 94: should have distance after complex ‘than that of 2 (2.066(8) Å)’
- Line 96 – use acronym PB but don’t define after first use of pentagonal-bipyramid. Please add.
- Authors should provide references for SHAPE programme used to determine continuous shape measure values. Please add: Inorg. Chem., 1998, 37, 5575–5582; Chem. Rev., 2015, 115, 13447–13483.
- Line 121: assuming the dc measurements were performed when lowering the temperature, should be written 300 – 2 K.
- Lined 153-154: The authors say ‘It also shows that the relaxation times in Orbach region for 1 with shorter Dy-O bond length in axial is always shorter than that of 2 at the same temperature’. However, the Dy-O bond lengths are longer for 1 than 2, so this should be corrected or clarified.
- In the ac susceptibility section, the authors should add a comment on the fact that the differences in the coordination environment as well as the crystal lattice/packing should result in differences in the spin-lattice relaxation processes. A recent review on such relaxation processes could be added to further this point. I recommend Dalton Trans., 2020,49, 9916-9928.
- Line 149: ‘These parameters are very similar to other D5h SMMs due to their sharing the same coordination geometry.’ – authors should provide references after this claim eg. Chem. Commun., 2020,56, 12037-12040.
Author Response
Referee 1:
The paper by Yan-Zhen Zheng and co-workers provides a thorough investigation of a new complex [Dy(OtBu)2(4-PhPy)5][BPh4] (1) with single-molecule magnet (SMM) behaviour, and draws direct comparisons to their previously reported compound [Dy(OtBu)2(4-PhPy)4][BPh4] (2). Although the synthetic procedures are identical apart from the solvent employed (toluene for 1 and THF for 2), this difference results in changes in the coordination environment around the Dy centre (7-coordinate for 1 and 6-coordinate for 2), despite the solvent being non-coordinating. This change in coordination number causes a change in symmetry around the Dy centre, which in turn has consequences on the magnetic properties. For 1, where more ligands are in the equatorial plane, symmetry is lowered and thus the barrier to relaxation is smaller than that of 2, which possesses higher symmetry around the Dy centre. Through structural and detailed magnetic investigations, supported by ab initio calculations, Yan-Zhen Zheng and co-workers draw on magneto-structural correlations to explain the differences in the SMM properties of 1 and 2. They also make use of theoretical calculations to determine the role of the solvent in determining the coordination environment, which was postulated to arise due to the differences in the electrostatic potentials of THF versus toluene, and as a result how these interact with the 4-PhPy ligands which reside in the equatorial positions. The review is very thorough and well written; it will certainly be of interest to the journals readers. However, I have a few minor suggestions to improve the manuscript.
Reply: We are grateful that the referee appreciates our work.
Q1: In abstract the authors write ‘More interestingly, this process is reversible and the magnetic energy barrier of 2 is higher than that of 1; while the relation of blocking temperature is just on the contrary due to the symmetry effect.’ I am not sure in which way the coordination number change is reversible, so this should be clarified in the main text.
Reply: The sentence has been revised.
Q2: The environment around the Dy centres is not strict in terms of symmetries, and so it is more correct to be referred to as pseudo-D5h and D4h or add a sentence in the crystallography section to explain that the symmetry is approximate and not strictly D5h and D4h.
Reply: We agree. But there are already too many claimed local D5h or other symmetries such as D4h and D6h which are actually pseudo only. We consider this is already a default setting in the community. It is absolutely true that any ideal symmetric symbol does not exist in real molecules.
Q3: The authors refer a lot to their previous report of complex 2, but don’t provide a reference for this other in the introduction where it is included in reference to previous Dy SMMs along with multiple others. It would certainly help the reader if this reference could be added where previous data (e.g. crystallography and magnetic properties) is referred to.
Reply: We gratefully thank the referee for reminding this. We have added the relevant references.
Q4: Sentence starting line 88: ‘The X-ray single crystal structure analysis reveals that the two complexes respectively compose of mononuclear seven-coordinate cation with [Dy(OtBu)2(4-PhPy)5]+ for 1 and six-coordinate cation with [Dy(OtBu)2(4-PhPy)4]+ for 2 motif’. The way this is phrased implies both complexes are seven coordinate. I would suggest the authors rephrase along the lines of ‘The X-ray single crystal structure analysis reveals that complex 1 is composed as a mononuclear mononuclear seven-coordinate cation with [Dy(OtBu)2(4-PhPy)5]+, whereas complex 2 has a six-coordinate cation motif [Dy(OtBu)2(4-PhPy)4]+’.
Reply: We have revised the sentences in the main text.
Q5: Line 94: should have distance after complex ‘than that of 2 (2.066(8) Å)’.
Reply: We have revised it.
Q6: Line 96 – use acronym PB but don’t define after first use of pentagonal-bipyramid. Please add.
Reply: We have added the definition of PB in the first use as you said.
Q7: Authors should provide references for SHAPE programme used to determine continuous shape measure values. Please add: Inorg. Chem., 1998, 37, 5575–5582; Chem. Rev., 2015, 115, 13447–13483.
Reply: The reference has been added.
Q8: Line 121: assuming the dc measurements were performed when lowering the temperature, should be written 300–2 K.
Reply: We have now revised the writing 300–2 K as you said in the main text.
Q9: Lined 153-154: The authors say ‘It also shows that the relaxation times in Orbach region for 1 with shorter Dy-O bond length in axial is always shorter than that of 2 at the same temperature’. However, the Dy-O bond lengths are longer for 1 than 2, so this should be corrected or clarified.
Reply: Many thanks. We have corrected the description in the main text.
Q10: In the ac susceptibility section, the authors should add a comment on the fact that the differences in the coordination environment as well as the crystal lattice/packing should result in differences in the spin-lattice relaxation processes. A recent review on such relaxation processes could be added to further this point. I recommend Dalton Trans., 2020,49, 9916-9928.
Reply: We have added a comment on the fact that the differences in the coordination environment as well as the crystal lattice/packing should result in differences in the spin-lattice relaxation processes according to your recommended reference.
Q11: Line 149: ‘These parameters are very similar to other D5h SMMs due to their sharing the same coordination geometry.’ – authors should provide references after this claim eg. Chem. Commun., 2020,56,12037-12040.
Reply: We have provided the reference in the main text. Many thanks.

Reviewer 2 Report
The manuscript 1265762 “Reversibly Switching the Local Symmetry from D5h to D4h for Single-molecule Magnets by non-Coordinating Solvents” by Xia-Li Ding, Qian-Cheng Luo, Yuan-Qi Zhai, Qian Zhang, Lei Tian, Xinliang Zhang, Chao Ke, Xu-Feng Zhang, Yi Lv and Yan-Zhen Zheng describes the synthesis and study of the magnetic properties of a new complex with a pentagonally bipyramidal structure of the coordination polyhedron (D5h) with five 4-phenylpyridines in the equatorial plane. Previously, its relative was studied, which has only 4 similar ligands and has the D4h symmetry of the coordination node.
The authors argue that this symmetry tuning is due to the difference in the electrostatic potentials of the solvent molecules used in the synthesis of the complexes.
This explanation is doubtful, since there is a significant difference not only in the synthetic procedure, but also in the compositions of the obtained substances. The authors deliberately indicated the wrong composition of the studied earlier complex 2: Na{[Dy(OtBu)2(4-Phpy)4][BPh4]2}*2thf*hex, instead of: {[Dy(OtBu)2(4-Phpy)4][BPh4]2, therefore, reasoning about the difference in electrostatic potentials is speculation.
The manipulation of the results, even those already obtained earlier, is unethical, so I ask the editor to refuse the authors to publish the manuscript
Author Response
Reviewer: 2
The manuscript 1265762 “Reversibly Switching the Local Symmetry from D5h to D4h for Single-molecule Magnets by non-Coordinating Solvents” by Xia-Li Ding, Qian-Cheng Luo, Yuan-Qi Zhai, Qian Zhang, Lei Tian, Xinliang Zhang, Chao Ke, Xu-Feng Zhang, Yi Lv and Yan-Zhen Zheng describes the synthesis and study of the magnetic properties of a new complex with a pentagonally bipyramidal structure of the coordination polyhedron (D5h) with five 4-phenylpyridines in the equatorial plane. Previously, its relative was studied, which has only 4 similar ligands and has the D4h symmetry of the coordination node.
Q1: The authors argue that this symmetry tuning is due to the difference in the electrostatic potentials of the solvent molecules used in the synthesis of the complexes. This explanation is doubtful, since there is a significant difference not only in the synthetic procedure, but also in the compositions of the obtained substances. The authors deliberately indicated the wrong composition of the studied earlier complex 2: Na{[Dy(OtBu)2(4-Phpy)4][BPh4]2}*2thf*hex, instead of: {[Dy(OtBu)2(4-Phpy)4][BPh4]2, therefore, reasoning about the difference in electrostatic potentials is speculation. The manipulation of the results, even those already obtained earlier, is unethical, so I ask the editor to refuse the authors to publish the manuscript.
Reply: The referee is overreacted with a formula. We understand that the full formula for complex 2 is Na{[Dy(OtBu)2(4-Phpy)4][BPh4]2}∙2thf∙hex, but this does not change the fact that the central Dy(III) is six-coordinate cation motif [Dy(OtBu)2(4-PhPy)4]+ with local D4h symmetry. For the sake of clarity it is reasonable to omit the guest molecules for porous materials. But to avoid annoying readers like the referee we have now used the full formula for complex 2 throughout the main text.
As for the explanation of the solvent effect it is plausible. The structural alternation manifests that solvent has an important influence on the coordination geometry and spatial packing of the structures. Herein, we use theoretical calculations to determine the role of the solvent and found the differences in the electrostatic potentials may be the reason. We are not saying this is the only and sole reason. We wish the referee understands.

Reviewer 3 Report
This manuscript by Ding et al. describes the preparation of two different Dy complexes starting from the same ligands but changing the solvent in which the reaction is carried out. One of the two complexes has already been reported by the same authors but is described here in comparison with the new species, which shows interesting SMM properties. The paper is sufficiently innovative to grant publication in Inorganics, however the presentation quality should be drastically improved: several sentences contain grammar mistakes and typos.
I only have on question on the scientific side: on page 4, line 147, authors present the function for 1/tau that they used to fit data in figure 3 d). This function contains Orbach ad Raman terms. However, on the same page, lines 150-153 the authors mention a QTM regime that becomes dominant at low temperature. Did they also use a term to describe QTM in the fitting function? Also, the data they report in Figure 3 d) contains high and medium temperature ranges (300-40 K) and no low temperature data is shown. QTM is not apparent in the reported data. Could the authors comment on this?
Author Response
Reviewer: 3
This manuscript by Ding et al. describes the preparation of two different Dy complexes starting from the same ligands but changing the solvent in which the reaction is carried out. One of the two complexes has already been reported by the same authors but is described here in comparison with the new species, which shows interesting SMM properties. The paper is sufficiently innovative to grant publication in Inorganics, however the presentation quality should be drastically improved: several sentences contain grammar mistakes and typos.
Reply: We gratefully thank the referee for appreciating our work and recommending it for publication in Inorganics, and we have now submitted the revised version with improved quality.
Q1: I only have on question on the scientific side: on page 4, line 147, authors present the function for 1/tau that they used to fit data in figure 3 d). This function contains Orbach and Raman terms. However, on the same page, lines 150-153 the authors mention a QTM regime that becomes dominant at low temperature. Did they also use a term to describe QTM in the fitting function? Also, the data they report in Figure 3 d) contains high and medium temperature ranges (100-40 K) and no low temperature data is shown. QTM is not apparent in the reported data. Could the authors comment on this?
Reply: Though the higher temperature relaxation mechanisms are similar in two complexes, in low temperature the process is disparate for two complexes. Due to the severe QTM existing in the low temperature region for 2 with local D4h symmetry effect we use a term to describe the QTM in the fitting function. While for 1 we did not use a term to describe the QTM. This is due to the absence of QTM effect at low temperature region for 1 which has local D5h symmetry. This can be also explained by the opening temperature of the hysteresis loops (Figure 3). At 2 K (Figure 3b) at the zero field region the hysteresis is much wider for 1 than that of 2. We have added comments on this.
